# Tracking Long-Lived Free Radicals in Dandelion Caused by Air Pollution Using Electron Paramagnetic Resonance Spectroscopy

**DOI:** 10.3390/molecules29215173

**Published:** 2024-10-31

**Authors:** Ireneusz Stefaniuk, Bogumił Cieniek, Agata Ćwik, Katarzyna Kluska, Idalia Kasprzyk

**Affiliations:** 1Institute of Materials Engineering, University of Rzeszow, Pigonia 1, 35-939 Rzeszow, Poland; bcieniek@ur.edu.pl; 2Institute of Agricultural Sciences, Land Management and Environmental Protection, University of Rzeszow, Zelwerowicza 4, 35-601 Rzeszow, Poland; acwik@ur.edu.pl; 3Institute of Biology, University of Rzeszow, Zelwerowicza 4, 35-601 Rzeszow, Poland; kborycka@ur.edu.pl (K.K.); ikasprzyk@ur.edu.pl (I.K.)

**Keywords:** persistent free radicals, EPR, particulate matter, *Taraxacum*, plants, stress

## Abstract

Studies on particulate air pollution indicate that a new type of pollutant should be considered from mainly fossil fuel combustion and automobile exhaust emissions, i.e., environmentally persistent free radicals. These radicals, ubiquitous in the environment, have a long life span and are capable of producing harmful reactive oxygen species. Samples of dandelion were collected in 2020 and 2021 in spring and late summer. Roots, leaves, flower stalks, and inflorescences of *Taraxacum* sp. were collected from six sites with three plants each, along with monitoring of particulate matter air pollution. Four sites were located at streets with heavy traffic and two were control sites in the rural part of the city. The free radical content in each part of the plant was measured by electron paramagnetic resonance. The leaf was selected as the most appropriate part of the plant for the measurement of carbon-derived free radicals. The g_eff_ value and the total number of spins were calculated. Relationships were found between location, season, and measurements. The electron paramagnetic resonance spectrum consists of at least two components, which can be attributed to C-type radicals and mixed C + O radicals. Their increase in numbers in the fall seasons, compared to the spring seasons, is also noticeable. It has also been observed that leaves collected in autumn have a higher g_eff_ value, which is probably related to the higher amount of oxygen- and carbon-derived free radicals.

## 1. Introduction

Particulate air pollution is one of the global environmental pollutants [1]. Particulate matter and the substances it contains are a strong stress factor for living organisms [2]. At the cell level, many stress factors, including pollutants, have common effects such as oxidative stress and the formation of harmful to cells reactive oxygen species (ROS), reactive nitrogen species (RNS), and reactive sulfur species (RSS) [3,4]. A detailed analysis of various dust fractions shows that the list of harmful components for living organisms is very long. These include organic and inorganic components of origin; for example, polycyclic aromatic hydrocarbons (PAHs), organic and elemental carbon, alkanes, organic acids, ROS, RNS, and heavy metals are considered key components of particulate matter (PM) that induce the generation of hydroxyl radicals (•OH) in Fenton-like reactions [5,6,7]. The recent studies indicate that a new type of pollutant, environmentally persistent free radicals (EPFRs), mainly from fossil fuel combustion and automobile exhaust emissions, should be added to this list [8,9,10,11]. Fossil fuel combustion, vehicle-related emissions, and industry are the primary sources of EPFRs, and they can be formed in the atmosphere as a result of oxidative reactions [10,12]. These are long-lived radicals that are ubiquitous in the environment, capable of producing harmful ROS species, which negatively affect the functioning of not only living organisms but also ecosystems [12,13,14,15]. Gehling et al. (2014) reported that one EPFR generates an average of 10 hydroxy radicals [16]. The radical content of plants can be investigated indirectly by methods based on the reduction of metal ions to ions of lower oxidation and the scavenging phenomenon of free stable radicals (FR) [17,18,19] or directly by FR content using the electron paramagnetic resonance method (EPR) [20]. Atmospheric particles may contain different types of EPFRs, such as semiquinone, phenoxyl, and other types of radicals [21,22,23,24]. The g-factor is a parameter used to distinguish between EPFR types [24,25]. Carbon-centered persistent free radicals generally have a g-factor of less than 2.003, carbon-centered radicals with adjacent oxygen atoms have a g-factor between 2.003 and 2.004 [26], and oxygen-centered persistent radicals generally gave a g-factor greater than 2.004. Soil organic matter is rich in semiquinone free radicals with g-factors between 2.0031 and 2.0050 [27]. Smoke and tar produced from the combustion of coal, petroleum, and tobacco contain large amounts of terpenoid free radicals with g-factors of approximately 2.0032 [28]. The g-factors of EPFRs in actual atmospheric particles range from 2.0030 to 2.0047 [29], which depends on the chemical composition and source of PM.

The EPR method shows high efficiency and sensitivity in the study of FRs in plants formed under the influence of air pollution. It is possible to measure the absolute number of spins and identify ROS forms; it has been described in many articles from which we have cited only a few [5,6,7,12,15,30]. The method does not require time-consuming preparation of material for analysis; the amount of material does not need to be large (a few milligrams).

In the literature, the EPR method was most often used to study the antioxidant capacity of dandelion using the 1,1-diphenyl-2-picryl-hydrazil (DPPH) radical [31,32]. With EPFRs, we study radicals with long lifetimes, in contrast to spin trapping methods in which the radical’s lifetime is very short.

In proceeding with the research presented here, it was assumed, as in [18], that the FR content of dandelions will be correlated with particulate air pollution. In areas with heavy traffic, particulate pollution will increase and the plants that grow there, responding to this stress factor, will manifest an increase in the FR content. In order to confirm this thesis, comprehensive studies were conducted at several sites taking into account traffic intensity and the structure of the urban tissue, referring to the city ventilation system and heating season. Being aware of the complexity of plant responses to external factors, including meteorological conditions, we undertook detailed analyses of FR content using the EPR method in all organs of one model plant in two seasons, which allowed us to obtain a full picture of the studied phenomenon. The recent studies by Vejerano and Ahn [30] show that the FRs present in leaves are also biogenic in nature. Due to the highest chlorophyll content and the intense photosynthetic process that occurs here, they are the place in the plant where the most FRs are naturally formed. The analysis of the number and type of EPFRs and comparisons of the FR content in different dandelion organs were used to confirm their thesis.

In studying the plant’s response to environmental stress, it is important to select an appropriate bioindicator. Dandelion is a plant often used for biomonitoring [18,33] because it meets the necessary criteria. It has a broad ecological optimum, is common and easy to identify and obtain, is resistant to contaminants, shows measurable responses at different sites, and with the same exposure to contaminants, should show similar and specific responses [34,35,36]. We chose dandelion as our study model for these reasons.

## 2. Results

### 2.1. Particulate Air Pollution

The sites were characterized by low concentrations of particulate matter, especially sites E and F, which were meant as a control. Although the measurement was temporary and the results cannot be related to the daily average critical values reported for these pollutants by the WHO, it can be assumed that the average concentrations of PM_10_ and PM_2.5_ at all sites were low. Exceedances of the WHO norms for PM_10_ did not occur and for PM_2.5_ were rare. The altitude at which the particles were measured was not significant. The time of year was a clearly differentiating parameter. Whatever the location of the test sites, the highest dust pollution (both fractions) was found in fall 2021. In spring 2020, dust concentrations were also not lower, although a lockdown was declared in March/April 2020 in Poland due to the COVID-19 pandemic and traffic was restricted (Figure 1A,B and Figure 2).

### 2.2. EPR Analyses of Dandelion

EPR spectra were measured for four parts of the plant (Figure 3). The observed line is a composite of at least two lines. Two components of the EPR line were extracted using EasySpin software (version 5.2.35). The most characteristic EPR spectra of each part from a single plant (17 April 2020, site No. D1) were selected for analysis. In the first step, the experimental spectrum was “fit” to two EPR lines. The results are shown in Table 1.

The A component is characterized by a much higher intensity in the EPR line than the B component. The component of the spectrum (A) with a value of g_eff_ < 2.003 is a signal from carbon from air pollutants (based on the literature [12]), while the other component (B) is from all other radicals of various origins (including C and C + O). The A lwpp values for the leaf are significantly higher than for the other parts of the plant. The total intensity of the A component corresponding to the dust-derived radicals (A weight) is 6.58 times higher than for the B component. For the other plant organs, the intensities of the two components are similar (Table 1). It was also found that for a leaf harvested in spring, the g_eff_ values are similar and significantly lower than for leaves harvested in autumn. We note that for the other organs, the proportion of A/B components = 1, while for the leaf, this ratio is 6.58. The example results of the EPR spectrum fit for the other parts of the plant are shown in Figure 4, Figure 5, Figure 6 and Figure 7 part (A).

The second step of the analysis was to simulate the full EPR spectrum with its components to verify the parameters of the EPR spectra obtained by fitting. Figure 4, Figure 5, Figure 6 and Figure 7 show the theoretical spectra for components A and B and the full theoretical spectrum.

Based on the EPR results obtained (Table 1) and described in the literature, a relationship was observed between the value of g_eff_ and the type of radical in a given part of the plant. Generally, values of the g_eff_ factor < 2.0030 are associated with C-centered radicals, the range 2.0030–2.0040 is associated with a mixture of O- and C-centered radicals, and values > 2.0040 are associated with O-centered radicals [7]. The fit results obtained indicate that component A is characterized by a value of g_eff_ < 2.003, which indicates a radical C, while component B is associated with oxygen and carbon radicals. In the root, flower stalk, and inflorescence, C and O-type radicals are equally present, while in the leaf, C-type radicals strongly dominate.

Measurements of the absolute number of spins were made for all parts of the plants for each location and season. The leaves had the highest FR content of 1.8 × 10^15^ spin/g. Inflorescences were the second most abundant organ in terms of FRs at 5 × 10^14^ spin/g.

Analyzing the results for the root and the flower stalk, we note slight differences with a slight trend indicating a higher value of radicals measured in autumn in both series. However, for inflorescence and leaf, the differences are much larger and indicate a much greater increase in the number of radicals in autumn. Additionally, we note that in autumn 2020, the highest radical values of the entire range of tests were obtained (Figure 1C and Figure 8).

We note that in the case of the root, there is practically no correlation; for the stalk, we can see a slightly higher value for position B and D compared to the others. For the inflorescence, we note that site C has the highest value of EPFRs, while the results for B and D may have more errors due to the smaller number of flowers included in the study. The EPFR results for the leaf also show the highest value for site C, followed by B and D, with the lowest value for site A (Figure 1C and Figure 8).

A detailed analysis of the mean values of g_eff_ was performed for leaves (see Figure 1D and Figure 9). The obtained values show large differences. The linear analyses presented in Table 1 reveal that both C and O radicals are present in the leaves, but the proportion of the former is much higher. Furthermore, as g_eff_ values increase, the number of oxygen radicals increases, the origin of which can be linked to EPFRs and other environmental factors.

Analyzing the results of EPFR measurements taking into account the number of spins, the g_eff_ value, and air pollution, we notice regularities. For autumn 2020, the largest g_eff_ values were obtained (an increase in O radicals), an increase in radicals compared to spring 2020. The air pollution measurements for spring and autumn 2021 agree quite well with the EPFR measurements for inflorescence. This can be explained by the relatively short lifespan of the inflorescence compared to the leaf, and for air pollution measurements on the day the sample is collected, the flower is most suitable for short-term pollution measurements. For long-term averaged measurements of air pollution, the leaf is be the best. Measurements of environmental factors, averaged air temperatures and average temperatures near the ground, and sunshine were carried out, and the results are shown in Figure 10 and Figure 11. We found a positive relation between climatic parameters such as average temperature, temperature at ground level, sunshine (only 2021 seasons), and the total number of fragrant radicals.

## 3. Discussion

Free radicals are ubiquitous in nature. They are formed under the influence of solar radiation, UV radiation, ionizing radiation, and electrical discharges. However, combustion processes (of anthropogenic and natural origin) and traffic-related pollution are sources of persistent long-lived free radicals (EPFRs). The study of FR content, in particular EPFR, by EPR is not very common and comprehensive, as Xu et al. highlighted in their literature review [12]. High values (10^18^–10^20^ spin/g) are recorded in, for example, biocarbons or particulate matter. Plants under stress are characterized by an elevated FR content (10^16^–10^18^ spin/g) [9,10,12,14]. Vejerano and Ahn [30] estimated the content of persistent biogenic free radicals at the level of 10^15^–10^16^ spin/g while demonstrating that the needles contained less FR than the leaves of broadleaf trees. The comparison of the data presented in the literature with those obtained from the study indicates that the number of FRs present in various dandelion organs at 10^14^–10^15^ spin/g is a low value.

The results obtained are interesting not because of their values, as mentioned above, but because of their diversity. The observed differences between sites, seasons, and organs prompt the search for probable causes. Although the sites where the plants were collected differed in terms of PM pollution and traffic intensity, no simple link was shown between the content in FRs and these variables. However, the location in the structure of the urban tissue conducive to ventilation seems clear. Leaves collected from the most trafficked site (A), located at the southern end of the city (Figure 12), contained relatively low levels of FRs, whereas site C, located furthest north with much lower trafficking (Figure 12), had high levels of FRs. It seems that the city ventilation factor is much more important than traffic pollution. In Rzeszow, the ventilation from the southwest sector (from S to W) is dominant and affects the blowing of pollutants from south to north. The important factor is also urban tissue associated with the possibility of ventilating the area. Site B is located within a compact urban fabric (Figure 12) conducive to the accumulation of pollutants, hence, the stronger response of plants growing here to stress than at site D (Figure 12), where, despite the highest traffic load, due to the open space, pollutants can be blown in the northern and eastern directions.

Reference sites E and F lie away from the main traffic routes, but the number of FRs, including those of carbon origin recorded in the spring, was not at all the lowest. Other causes should be sought. During the heating season (spring campaigns 2020 and 2021), low emission pollution from single-family houses located in close proximity to site E could have been a serious stress factor. Air pollution from residential heating is a significant problem in this region. Even during the 3-week lockdown in 2020 at the turn of March and April due to the COVID-19 pandemic, PM pollution was not significantly lower in the city, although this has been confirmed in other regions of the world [5]. In temperate climates (see Figure 10 and Figure 11), apartments must be heated at that time, which is not conducive to good air quality. In China, it was found that the decrease in PM_2.5_ concentration during the COVID-19 lockdown did not reduce its toxicity because its oxidation potential increased, and the main identified factors influencing it were secondary emissions and coal combustion [5]. A detailed analysis of the obtained results indicates that season is a clearly discriminating factor in the results, but surprisingly, it is not related to air quality and PM concentrations as we assumed when undertaking the research. At each site, the number of FRs was higher during the measurements in early autumn, when in addition to radicals of carbon origin, we find those of oxygen origin. Climate seems to be an important factor. Temperature stress and atmospheric photochemical processes may be the cause of their formation [10]. Insolation, UV radiation, and ozone in the air positively correlate with the radical concentrations [16]. In the study area in September 2020 and 2021, the air and ground temperatures were clearly higher than in spring (see Figure 10 and Figure 11); the sun then operates strongly and longer.

The study confirms reports by other authors [12,30] that the leaf is the organ where the content of FRs is high and of various origins. Photosynthesis and respiration, which occur most intensively in the leaf, produce FRs of biogenic origin [12,30,38]. Furthermore, leaves live longer than other organs and are more exposed to pollution and long-term exposure to sun, which causes the formation of FRs. As a result of the analysis, two components of the EPR spectra were obtained; the first (A), with a g_eff_ value < 2.003, is a signal of carbon from air pollution (based on the literature [5,7]), while the second component comes from other radicals mainly of oxygen O and carbon C origin. For the parts of the plant, the A/B is 1 (Table 1), while in the leaf, we observe more than six times more C-type radicals associated with EPFRs than the others. It was also observed that for the EPR spectra of the leaf in spring, we have a lower value of g_eff_, while in autumn, we observe an increase; probably this is related to the higher amount of oxygen O and carbon C free radicals. The FR concentrations in tree leaves [9] are similar to those in dandelion leaves and one order of magnitude higher. Thus, our research confirms the reports of other authors [9] that the leaf is the organ in which the content of free radicals is high and has various origins, both exogenous and endogenous.

## 4. Materials and Methods

### 4.1. Study Area

The study was carried out in 2020 and 2021 in Rzeszów (SE Poland) during two seasons: heating season (spring) and non-heating (autumn). The axis of the city is the Wisłok River, which organizes both the urban fabric and the main traffic arteries that run from SW to NE. Such a course of the river and the main communication arteries makes it easier to ventilate the city, because the wind directions from the SW sector are one of the most common in the city. Taking into account this assumption, a high-traffic communication route running through the entire city from SW to NE and then to N in line with the expected prevailing wind direction was chosen for the location of the research sites (A–D). Therefore, it can be expected that areas located near the southern sections of roads will be characterized by less pollution (site A) than those located inside the dense urban tissue (sites B and D) or in the north (site C). Sites E and F had a control character and were located outside of this transect, in a rural area away from the streets (Figure 12).

The origin of particulate matter and other harmful substances in the environment can be influenced by the amount of automobile traffic. Another assumption we made, therefore, was that plants at sites with increased car traffic (A and D) would show increased symptoms of stress.

### 4.2. Materials

The model organism in the study was a perennial plant *Taraxacum* sect. *Taraxacum* is considered a good bioindicator of air pollution [18,33,35,36,39]. The material used for the study was roots, leaves, flower stalks, and inflorescences. They were collected during 2 spring and 2 fall campaigns (17 April 2020, 17 September 2020, 12 May 2021, 15 September 2021). At each site, 3 plants growing relatively close to each other were sampled each time. Each sample was packaged separately, labeled, then transferred to the laboratory and stored in the freezer until measurement.

### 4.3. Measurement of Air Pollution

During the sampling of plant materials, the air pollution of particulate matter with PM_10_ and PM_2.5_ fractions was measured using a DustTrak II analyzer (model 8532, TSI Incorporated, Shoreview, MN, USA). The measurements were carried out at ground level, where dandelions grow, and at a height of about 1.5 m, which corresponds to the average height of the human nose and is similar to the height at which constant monitoring of pollution is carried out by state administration services. In addition, basic meteorological parameters were measured at the place where the biological material was collected: temperature, humidity, and wind speed. The air measurements were carried out between 12 a.m. and 3 p.m. Only sunny days with no rain and relatively high temperature were chosen as the proper time for data sampling.

### 4.4. Electron Paramagnetic Resonance

The collected material was dried at room temperature before EPR measurements; the removal of water significantly improves the sensitivity and tuning of the EPR spectrometer. Then, 20 mg of each sample was weighed in a quartz capillary, which was placed in the resonance chamber of the spectrometer. For measurements, a Bruker FT-EPR ELEXSYS E580 spectrometer (Bruker Analytische Messtechnik, Rheinstetten, Germany) was used. The spectrometer operated at X-band (~9.4 GHz). The following settings were used: central field, 3351.00 G; modulation amplitude, 1 G; modulation frequency, 100 kHz; microwave power, 94.64 mW; power attenuation, 2.0 dB; scan range, 80 G; conversion time, 25 ms; and sweep time, 25.6 s. The spectra were recorded in 1024 channels using the Xepr 2.6b.74 software. The signal was integrated twice to determine its area and concentration of the radicals.

## 5. Conclusions

The pollution analysis showed that the impact of traffic is less significant than the air movement associated with the city’s ventilation system. A detailed analysis shows that it is the season and climatic conditions rather than air quality that differentiate the results. We confirm the research of other authors that the leaf is an organ in which the FR content is high and of different origin; for this reason, the leaf illustrates well the average pollution during its life. To obtain information about pollutants in the environment from a shorter period, a shorter-lived organ, such as an inflorescence, should be selected. The ERP method we chose seems to be an excellent tool for assessing pollutants in the environment and the response of plants to these pollutants due to the fact that it allows us to identify the type of radical and the total level of pollutants: EPFRs and biogenic free radicals).

## Figures and Tables

**Figure 1 molecules-29-05173-f001:**
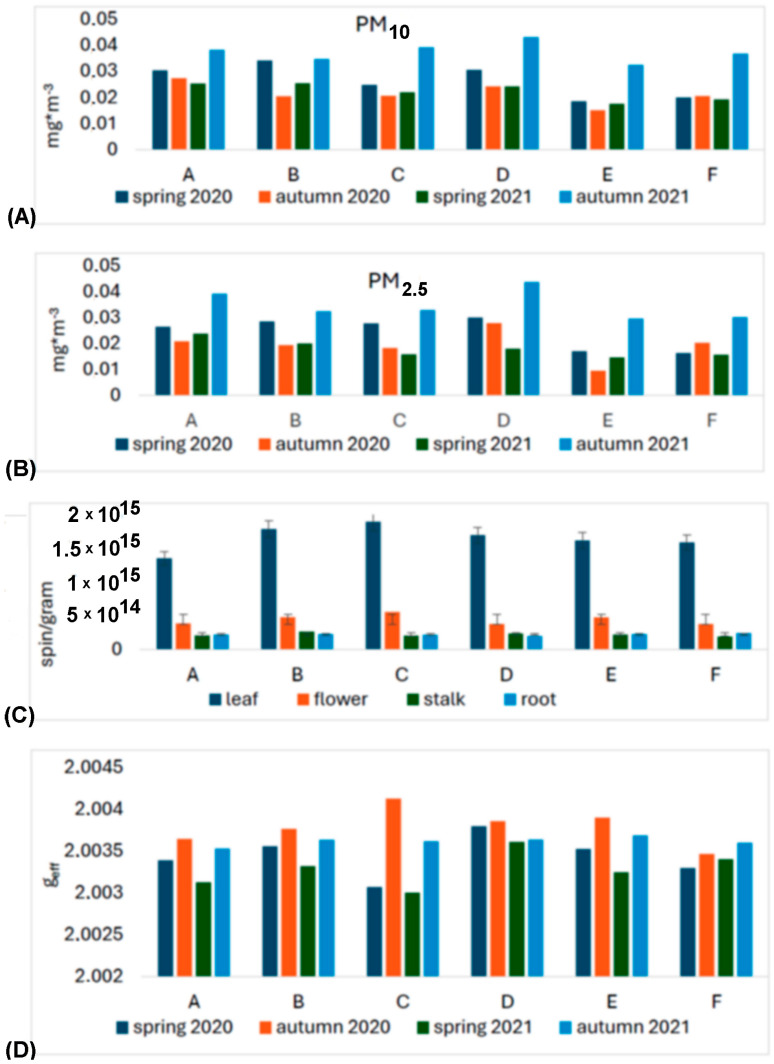
(**A**) Values of average air pollution PM_10_ and (**B**) PM_2.5_ measured at two heights on the days of collecting plant material; (**C**) values of spin/g count for the dandelion organs depending on the site; (**D**) mean g_eff_ values for the leaf for the site and collection periods (sites: A—Podkarpacka; B—Dabrowskiego; C—Warszawska; D—Śreniawitów; E—Staroniwa; F—Strzelnica).

**Figure 2 molecules-29-05173-f002:**
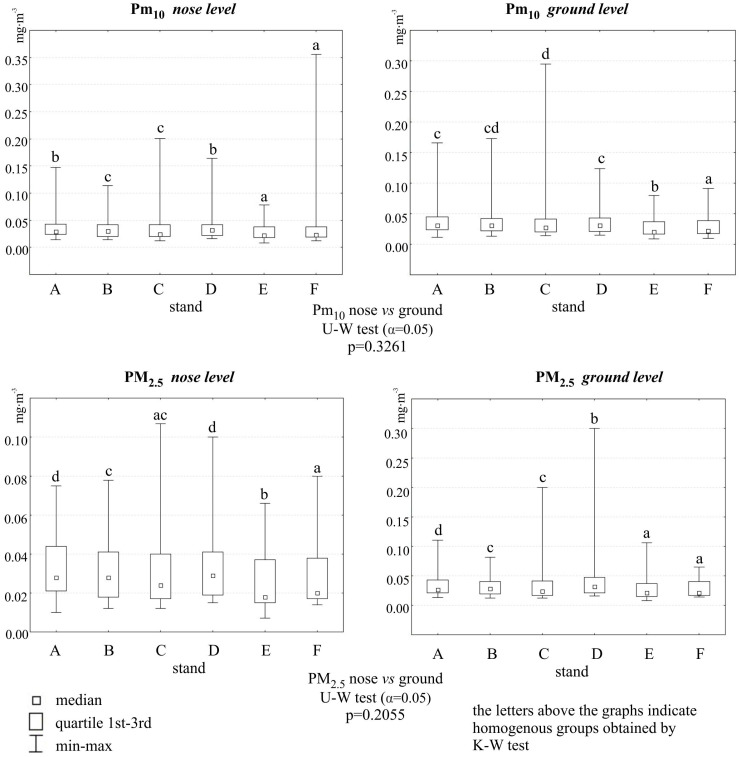
Concentrations of PM_10_ and PM_2.5_ [mg·m^−3^] at two heights at six sites., street symbols: A—Podkarpacka; B—Dabrowskiego; C—Warszawska; D—Śreniawitów; E—Staroniwa; F—Strzelnica. (letters a, b, c, d indicate the homogenous groups distinguished by Kruskal–Wallis post-hoc test).

**Figure 3 molecules-29-05173-f003:**
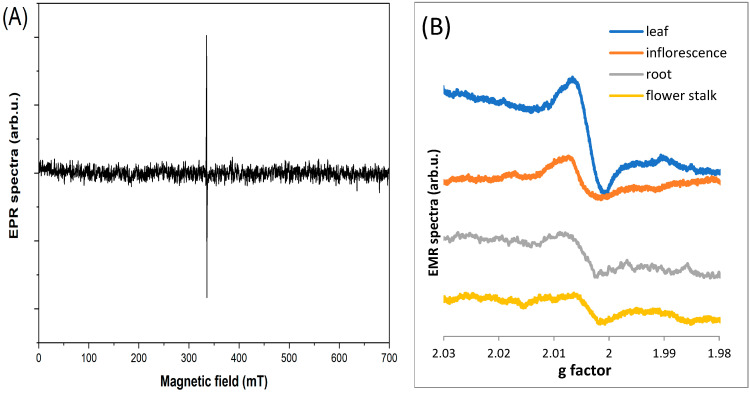
Example of an EPR spectrum, in full spectrum (**A**); The g factor dependences of EPR spectra for different parts of the plant, root, flower stalk, inflorescence, and leaf (**B**).

**Figure 4 molecules-29-05173-f004:**
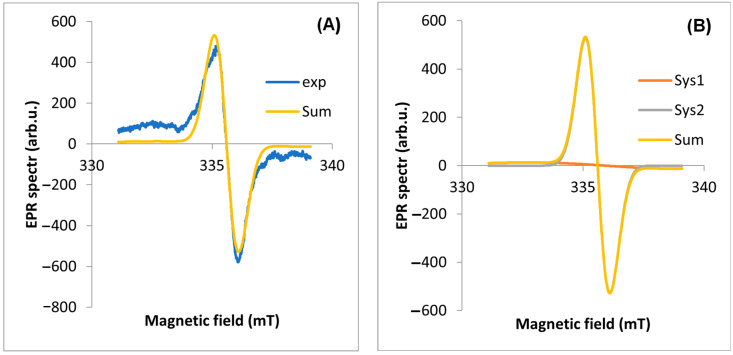
EPR spectrum for leaf: (**A**) exp—experimental spectra, Sum—theoretical spectra, (**B**) Sys1—EPR spectrum for first component, Sys2—EPR spectrum for the second component of the EPR spectrum fitted using the EasySpin software.

**Figure 5 molecules-29-05173-f005:**
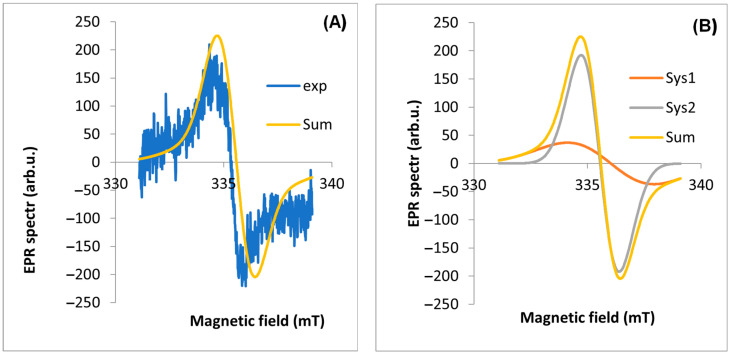
EPR spectrum for flower stalk: (**A**) exp—experimental spectra, Sum—theoretical spectra, (**B**) Sys1—EPR spectrum for first component, Sys2—EPR spectrum for the second component of the EPR spectrum fitted using the EasySpin software.

**Figure 6 molecules-29-05173-f006:**
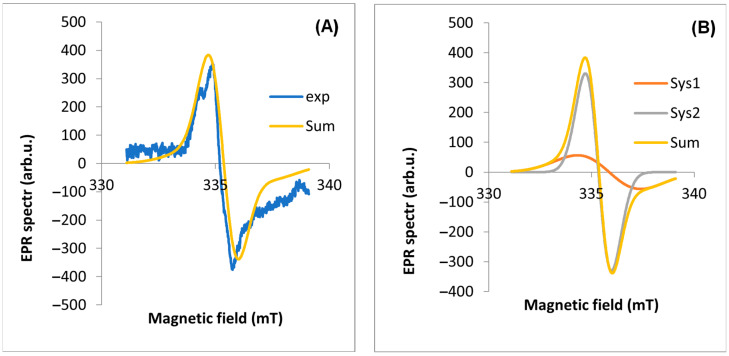
EPR spectrum for inflorescence: (**A**) exp—experimental spectra, Sum—theoretical spectra, (**B**) Sys1—EPR spectrum for first component, Sys2—EPR spectrum for the second component of the EPR spectrum fitted using the EasySpin software.

**Figure 7 molecules-29-05173-f007:**
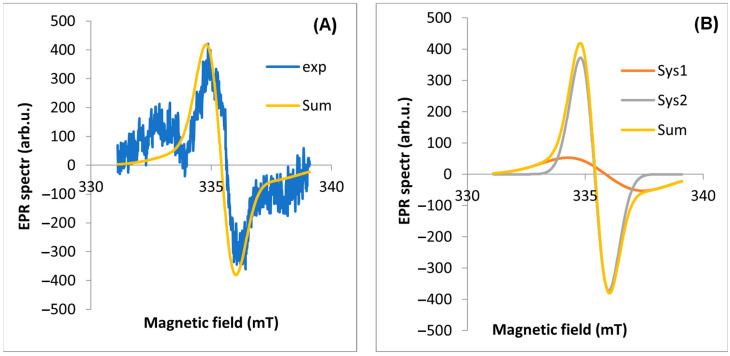
EPR spectrum for root: (**A**) exp—experimental spectra, Sum—theoretical spectra, (**B**) Sys1—EPR spectrum for first component, Sys2—EPR spectrum for the second component of the EPR spectrum fitted using the EasySpin software.

**Figure 8 molecules-29-05173-f008:**
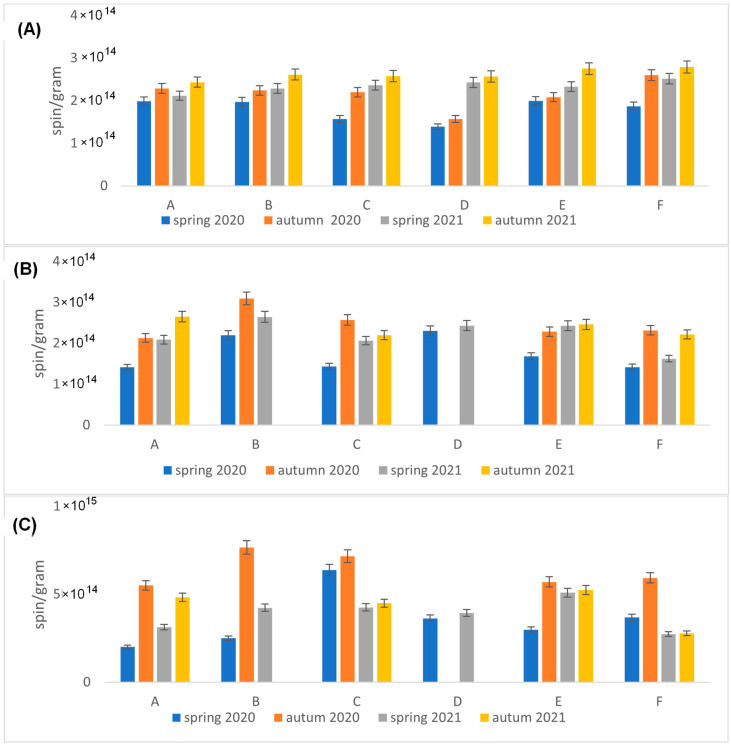
Values of spin/g count for the dandelion organs (**A**) root, (**B**) stalk, (**C**) inflorescence, (**D**) leaf, depending on the site and season; in the autumn campaigns, it was not possible to collect stalks and inflorescence from all locations).

**Figure 9 molecules-29-05173-f009:**
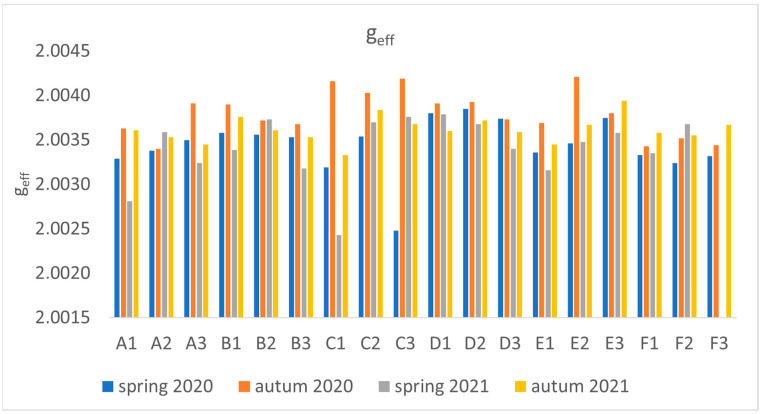
Effective g values for leaf depending on the location (symbols A–F) and season. Numbers represent the plant number.

**Figure 10 molecules-29-05173-f010:**
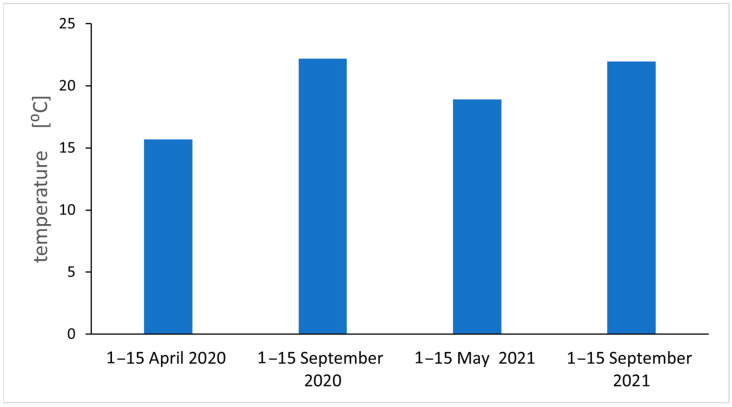
Average maximum air temperature in individual seasons.

**Figure 11 molecules-29-05173-f011:**
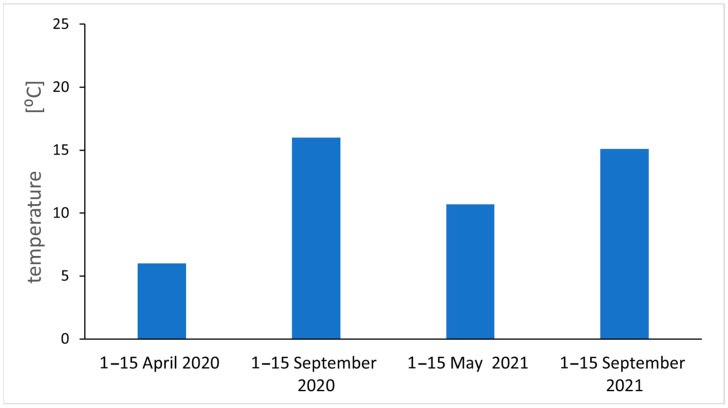
Average maximum air temperature at ground level in individual seasons.

**Figure 12 molecules-29-05173-f012:**
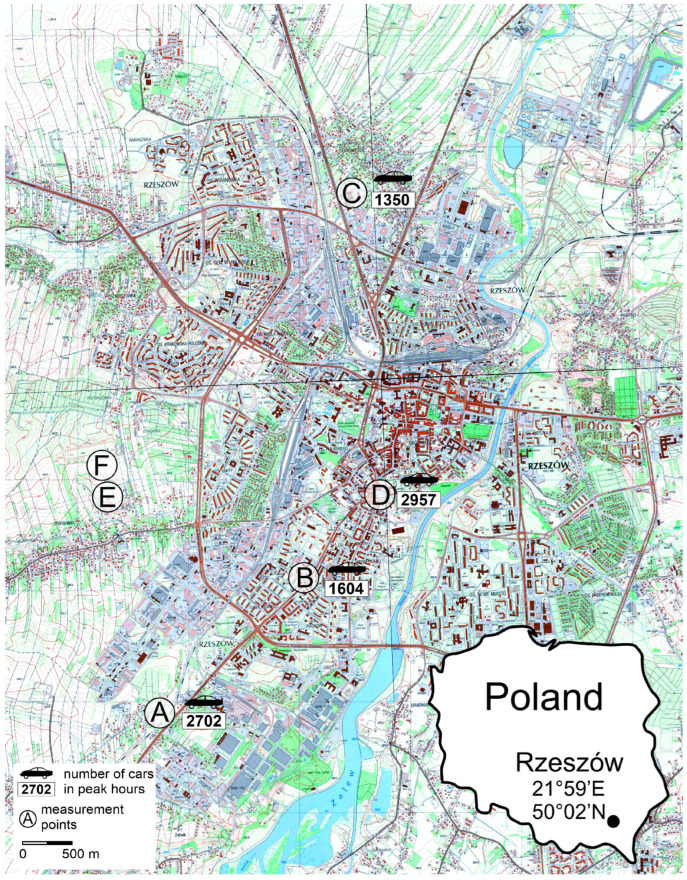
Location of the study area, street symbols: A—Podkarpacka; B—Dabrowskiego; C—Warszawska; D—Śreniawitów; E—Staroniwa; F—Strzelnica. Number of cars—the average value of the number of conventional vehicles per hour, calculated from the two peak hours of traffic in the Rzeszów Functional Area, i.e., 7–8 a.m. and 3–4 p.m.; calculated on the basis of data included in the report: Raport częściowy nr 4. Badania natężenia ruchu drogowego. Politechnika Rzeszowska, Rzeszow, Poland (2023) [37].

**Table 1 molecules-29-05173-t001:** Parameters of the experimental “fit” of the spectrum of two EPR lines: A—parameters for the first component of the EPR spectrum, B—parameters for the second component, compared with the experimental data for one plant from site D1.

EPR Line Components		Inflorescence	Stalk	Root	Leaf
Experimental line	Intensity [arb. unit]	0.223	0.112	0.124	1.332
Experimental line	g_eff_	2.0040	2.0038	2.0039	2.0038
A	g	2.0025	2.0015	2.0025	2.0021
A	lwpp [mT]	3.0804	3.8110	3.1860	6.3861
A	weight	1	1	1	6.58
B	g	2.0056	2.0043	2.0052	2.0040
B	lwpp [mT]	1.2740	1.6700	1.2000	1.0082
B	weight	1	1	1	1

g_eff_—effective spectroscopic splitting factor, lwpp—peak-to-peak line width, value A weight = 6.58 means that the total intensity of this component is 6.58 times higher than for the component B weight = 1.

## Data Availability

The dataset is available on request from the authors.

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
