# Peer review of "Tracking Long-Lived Free Radicals in Dandelion Caused by Air Pollution Using Electron Paramagnetic Resonance Spectroscopy"

_molecules, 2024, doi:10.3390/molecules29215173_

Round 1

Reviewer 1 Report

Comments and Suggestions for Authors

In this work, the authors used dandelion as a sample and studied the relationship between location, season, and measurement by monitoring PM air pollution and measuring the environmental persistent free radical content in various parts of dandelion. However, the introduction of the research background is not comprehensive, and the environmental significance of the research results is not clear. Therefore, I suggest that the study be reconsidered after major revisions. The detailed comments are as follows:

1. The introduction section provides less information on the current research progress in environmental persistent radicals.

2. It is necessary to indicate in the caption of Figure 1 what the letters ABCDE represent.

3. The selection of two sampling heights in the sampling process is based on what? Why not choose the sampling height related to dandelion plant growth height? Please explain the reason.

4. What do the lowercase letters a, b, c, etc. in Figure 2 represent? There are no annotations in the text.

5. What is the scientific significance of the research conclusion? It only confirms the reports of other authors and does not propose new scientific discoveries. Whether the conclusions of this study are universal. Please explain in the paper what specific scientific problems have been solved.

6. EPR spectra of dandelions from different seasons and locations need to be provided.

7. Some data in Figure 8b and c are not shown in the figures.

8. The conclusion mentions that the influence of urban ventilation factors needs to consider the different wind directions in different seasons.

Author Response

In this work, the authors used dandelion as a sample and studied the relationship between location, season, and measurement by monitoring PM air pollution and measuring the environmental persistent free radical content in various parts of dandelion. However, the introduction of the research background is not comprehensive, and the environmental significance of the research results is not clear. Therefore, I suggest that the study be reconsidered after major revisions. The detailed comments are as follows:

  1. The introduction section provides less information on the current research progress in environmental persistent radicals.

Ad. 1. The introduction has been revised and expanded. The paragraph describing and supporting the use of the EPR method to study EPFR radicals has been expanded. Environmental concerns have also been expanded.

  1. It is necessary to indicate in the caption of Figure 1 what the letters ABCDE represent.

Ad.2.  It was corrected: A – Podkarpacka; B- Dąbrowskiego; C- Warszawska; D- Śreniawitów; E – Staroniwa; F- Strzelnicza – names of streets in Rzeszów.

  1. The selection of two sampling heights in the sampling process is based on what? Why not choose the sampling height related to dandelion plant growth height? Please explain the reason.

Ad. 3. Two sampling heights of air pollutants (ground and nose level) have been chosen because they are related to:

- ground level- dandelion plant growth height - dandelions grow directly near the ground.

- the average height of human nose- that reflect level of pollution important from the inhabitant point of view (indicate which city districts are less or more pollutant); continuous monitoring of air pollution carried out by the state administration services is also carried out at a similar height.

In this way, we obtained more complete information on air quality. The PM drift in the air and they concentrations level might be different despite little difference of height measurement.

We have updated Chapter 4.2.1 with this information.

  1. What do the lowercase letters a, b, c, etc. represents in Figure 2 represent? There are no annotations in the text.

Ad. 4.  Letters a, b, c, d in fig. 2. are related to homogenous groups divided by K-W post-hoc test.

  1. What is the scientific significance of the research conclusion? It only confirms the reports of other authors is provided and does not propose new scientific discoveries. Whether the conclusions of this study are universal. Please explain in the paper what specific scientific problems have been solved.

Ad. 5 We agree with the reviewer that in the first version of the article these elements were not properly presented. Therefore, we have now detailed the conclusions of the study, which we have included in a separate chapter: 5 Conclusions. The results we obtained can contribute to the development of recommendations for urban planning and especially to the discussion of the planning and arrangement of important transportation routes. Therefore, the work presented is a comprehensive approach to studying the response of plants to both natural and anthropogenic factors. We believe that the results obtained have universal character as they relate to a wide range of factors. We emphasize that the study of the content of FR and EPRF using the EPR method is not common in the literature. This is because there are no works in which a single model organism was studied in such detail as we did. There are works in the literature that describe the antioxidant capacities of different parts of the plant, but not on the measurement of persistent radicals whose source is EPFRs. The EPR method that we used allows for interdisciplinary studies of air pollution from different sources using living plants.

  1. EPR spectra of dandelions from different seasons and locations need to be provided.

Ad. 6. Presenting all EPR spectra is problematic due to the fact that we have 6 sites with 3 plants and 4 plant parts each, in the study for 4 seasons. We obtain a total of 288 EPR spectra, so we have included only selected spectra, while the final results from the calculation in the form of spins/g, season, and location dependence are shown in Figures 8 and 9. We have also included an example EPR spectrum (Figure 3a) of a leaf in the full range of the X-band magnetic field.

  1. Some data in Figure 8b and c are not shown in the figures.

Ad. 7. When constructing the experiment, we assumed that the material would be collected at the same sites in the spring and fall campaigns, when dandelions are in bloom. Unfortunately, in autumn it was not possible to find inflorescences at all sites, hence the lack of data. We have taken this into account in the caption of the figure.

  1. The conclusion mentions that the influence of urban ventilation factors needs to consider the different wind directions in different seasons.

Ad. 8. We have expanded the 4.1 subsection of the detailed explanation of the choice of investigated sites in the context of urban ventilation. In light of our research, it seems that plants respond to stress with some delay (highest rates in autumn). Moreover, the location of our sites in relation to the main wind directions and their traffic load indicates that it is not momentary local factors (such as exhaust emissions, weather conditions) that affect a plant's response to stress, but the factor(s) that act over the long term. The long-term effect of negative factors is influenced by positioning within the urban fabric or the prevailing wind direction. Our study showed such correlations and confirmed the validity of this choice of variables. Therefore, we did not study the correlation between the prevailing wind direction in a given season, especially since we could not compare this type of data with the traffic load, since it is calculated as an annual average.

Reviewer 2 Report

Comments and Suggestions for Authors

The present manuscript describes an EPR study on plants as a useful method for assessing the pollution in various locations of the city of Rzeszów (Poland). The starting hypothesis and the period covered by the study, that of the COVID pandemic, are interesting. However, in my opinion there are major items that need to be clarified before publication.

The first question is why to use this method to study particulate air pollition instead of pollution caused by gases. One could expect that the influence of pollution particles on the roots, for instance, should be low. In fact, at first glance there seems to be no relationship between the number of particles and the number of radicals measured. On the contrary, the diffusivity of gases and the interplay between them and ambient humidity could affect all the parts in the plant.

Initially, the Authors seem to assume that most RFs come from pollution, although they later qualify this idea mainly when analyzing the results obtained in the leaves. Have the Authors collected any dandelions sample harvested in natural parks or areas far from towns, traffic and factories? It would be quite interesting to check the actual influence of endogenous radicals in leaves. Following with the implications, the relationship with the analysis of environmental conditions and studies of particle quantity measurements in different areas is crucial. In this regard, Authors mention that measurements of environmental factors are given in the supplement. However, apart from data given in Figures 1, 2, 10 and 11, this reviewer has not been able to access this information. These results should be given, at least as reviewing purposes and Supporting Information.

In my regard, the comparison among the different places is not so easy as it is given in the last paragraph (page 8, lines 151-156). As an example, note in the case of the stalk the high values of the signals detected in the samples harvested in B (Figure 8b).

The sampling period coincided with the COVID pandemic. As fas as this reviewer is aware, in March 2020 lockdown control measures were implemented in Poland. The policy measures drastically affected traffic and industrial activity. Have the authors analysed its influence and, accordingly, if the results they present can be actually extrapolated to ordinary years? In short, contextualizing the study within the framework of the COVID pandemic could be interesting.

From a more technical point of view, the procedures relating to sample preparation are not adequately described. For instance, it is not clear the number of plants measured, the number of measurements used to create the different figures. In fact, there is a lack of data from D in Figure 8 and it is not explained why. Were the plants allowed to air dry at room temperature for days? When or how was it estimated that the plants were sufficiently dry? As the Authors point out, the presence of water greatly affects the quality of the EPR measurements.

Some of the spectra, and the interpretations given, are not clear. Spectrum in Figure 6a suggests the presence of an unidentified signal with maximum at around 334 mT. Spectra in Figures 5a and 7a are difficult to be differenced from the background or cavity artifacts. See root and stem in Figure 3. Is there any influence of the minerals, mainly iron phases, present in the ground? Perhaps providing some spectra over the full range (e.g. 0-700 mT) would help explain this aspect.

Results show a certain degree of disagreement with those reported by Kang et al. Korean Journal of Food Preservation Vol. 9. No. 2, pp.253, about the antioxidative behaviour observed in leaves of dandelions due to their radical scavenger activity. However, in line with what is described in the present manuscript, it has been reported that the antioxidative power in the roots of dandelions is 10–12-fold higher than in the green parts of the plant (Jung et al. Spectrochimica Acta Part A 63 (2006) 846–850).

The writing must be carefully reviewed.

Some of the terms used may be common in environmental chemistry, but may not be so common to a general reader. Thus, abbreviations such as PM and PAH should be presented in text.

Some minor mistakes are given below:

Table 1: change G to g.

Line 169: whose origin of which

Line 184: shown in the supplement. Perhaps are the Figures 10 and 11?

Line 202: “Studies on the content of FR by EPR are not very common.” seems to repeat the previous sentence.

Line 207: change “then” to “than”.

Line 223: change “2000” to “2020”.

Lne 235: change “lives” to “live”.

Lines 244-245: please, revise the sentence.

Line 256: change “Figure 1” to “Figure 12”.

Line 257: please, revise the name of the species.

Line 275: please, revise the sentence.

Author Response

The present manuscript describes an EPR study on plants as a useful method for assessing the pollution in various locations of the city of Rzeszów (Poland). The starting hypothesis and the period covered by the study, that of the COVID pandemic, are interesting. However, in my opinion there are major items that need to be clarified before publication.

The first question is why to use this method to study particulate air pollition instead of pollution caused by gases. One could expect that the influence of pollution particles on roots, for instance, should be low. In fact, at first glance there seems to be no relationship between the number of particles and the number of radicals measured. On the contrary, the diffusivity of gases and the interplay between them and ambient humidity could affect all the parts in the plant.

Like many authors studying stable radicals of EPRFs by the EPR method, we also decided to analyze on the basis of g-factor values. We have added a paragraph in the introduction describing such studies. Moreover, for impurities of small size (smaller than the stomatal apparatus), there is the possibility that EPRFs can penetrate deep into the plant.

Initially, the Authors seem to assume that most RFs come from pollution, although they later qualify this idea mainly when analyzing the results obtained in the leaves. Have the Authors collected any dandelions sample harvested in natural parks or areas far from towns, traffic and factories?

The main idea was to check the FR content of different organs of the same plant collected at sites located along busy streets, which constitute a traffic route that runs through the entire city from south to north (sites A-D). Two sites E and F (without traffic) were our control. These are sites outside of the main urban fabric. Due to the fact that Rzeszow has doubled in size since 2005, the main part of its outskirts has an agricultural character, without a developed urban fabric. Sites E and F were located in just such an area, as can also be seen in Figure 12. Due to the comment, we expanded Section 4.1.

It would be interesting to check the actual influence of endogenous radicals in leaves. Following with the implications, the relationship with the analysis of environmental conditions and studies of particle quantity measurements in different areas is crucial. In this regard, Authors mention that measurements of environmental factors are given in the supplement. However, apart from data given in Figures 1, 2, 10 and 11, this reviewer has not been able to access this information. These results should be given, at least as reviewing purposes and Supporting Information.

We are very sorry, a minor error has sneaked in here. The environmental data are only in Figures 1, 2, 10 and 11, and we do not have a supplement for this manuscript.

In my regard, the comparison among the different places is not so easy as it is given in the last paragraph (page 8, lines 151-156). As an example, note in the case of the stalk the high values of the signals detected in the samples harvested in B (Figure 8b).

Yes, the reviewer rightly noted that the interpretation was not straightforward. The figures show the averaged values we used for comparisons, but sometimes we assisted with full data to clarify the conclusion.

The sampling period coincided with the COVID pandemic. As fas as this reviewer is aware, in March 2020 lockdown control measures were implemented in Poland. The policy measures drastically affected traffic and industrial activity. Have the authors analysed its influence and, accordingly, if the results they present can be actually extrapolated to ordinary years? In short, contextualizing the study within the context of the COVID pandemic could be interesting.

Indeed, the inspiration for carrying out this research was to take advantage of the reduced traffic situation of cars related to lockdown. We assumed that particulate air pollution would be significantly lower than in the previous year in March/April 2020. However, our research and analysis of the data presented by the air monitoring service did not confirm this. Yes, in 2020 the state services recorded slightly lower dust concentrations than in 2019 and 2021, but only in the second half of April. It seems that in Rzeszow the concentration of particulate matter at that time depended on emissions from heating stoves and boiler rooms. As suggested by the reviewer, we addressed this in Section 2.1 and discussed this in the Discussion section.

From a more technical point of view, the procedures relating to sample preparation are not adequately described. For instance, it is not clear the number of plants measured, the number of measurements used to create the different figures. In fact, there is a lack of data from D in Figure 8 and it is not explained why. Were the plants allowed to air dry at room temperature for days? When or how was it estimated that the plants were sufficiently dry? As the Authors point out, the presence of water greatly affects the quality of the EPR measurements.

The method of sample preparation is described in Sections 4.1 and 4.2.2. At each of the six sites, 3 plants were taken for testing. The samples were dried for several days at room temperature and protected from contamination, and the EPR measurement taken at 2 dB attenuation confirmed that the part of the plant was fully dried.

Some of the spectra, and the interpretations given, are not clear. Spectrum in Figure 6a suggests the presence of an unidentified signal with maximum at around 334 mT. Spectra in Figures 5a and 7a are difficult to differentiate from the background or cavity artifacts. See root and stem in Figure 3. Is there any influence of the minerals, mainly iron phases, present in the ground? Perhaps providing some spectra over the full range (e.g. 0-700 mT) would help explain this aspect.

The reviewer rightly pointed out that some EPR signals have low intensity, but for example, in Figure 7a the amplitude of the noise is about 100 units while the amplitude of the signal is close to 800 units. Thus, the signal-to-noise ratio is 8:1. Of course, we took into account other signal sources (e.g., iron ions) which we eliminated. The signal suggested by the reviewer at 334 mT is an artifact, since the other spectra from this location do not have this signal.

In the paper, we include a spectrum over a wider range of magnetic fields (Figure 1a), where the signal from the permanent radical is clearly visible.

Results show a certain degree of disagreement with those reported by Kang et al. Korean Journal of Food Preservation Vol. 9. No. 2, pp.253, about the antioxidative behaviour observed in leaves of dandelions due to their radical scavenger activity. However, in line with what is described in the present manuscript, it has been reported that the antioxidative power in the roots of dandelions is 10-12 times higher than in the green parts of the plant (Jung et al. Spectrochimica Acta Part A 63 (2006) 846–850).).

Thank you for pointing out articles in which dandelion measurements by EPR were described. In these papers, antioxidant capacity was studied using DPPH radical spin trapping. We have added information about such studies in the introduction. On the contrary, direct comparisons of the results from these works with ours are very complicated. Our measurements are related to persistent radicals and their identification is based on a comparison of g values (added paragraph in the Introduction); we do not study antioxidant capacity. Our results show the level of persistent radicals in different parts of the plant and identify their type.

The writing must be carefully reviewed.

Some of the terms used may be common in environmental chemistry, but may not be so common to a general reader. Thus, abbreviations such as PM and PAH should be presented in text.

It was explained in Abstract and in Introduction.

Some minor mistakes are given below:

Table 1: change G to g.

All mistakes were corrected

Line 169: whose origin of which

Line 184: shown in the supplement. Perhaps are the Figures 10 and 11?

Line 202: “Studies on the content of FR by EPR are not very common.” seems to repeat the previous sentence.

Line 207: change “then” to “than”.

Line 223: change “2000” to “2020”.

Lne 235: change “         ” to “live”.

Lines 244-245: please, revise the sentence.

Line 256: change “Figure 1” to “Figure 12”.

Line 257: please, revise the name of the species.

It was corrected. Latin names were written in Italic

Line 275: please, revise the sentence.

Round 2

Reviewer 1 Report

Comments and Suggestions for Authors

The quality of the manuscript has been greatly improved with revisions, the manuscript has more detailed data and some of the reviewers' concerns have been addressed, but the readability of the article remains a concern. From a scientific perspective, the method used in this article is a mature technical method, and the conclusions obtained are not obviously novel, but the readability of the article seriously affects the quality of the article, and what is more important is that the collected data are all from 3 or 4 years ago, which can be referenced at a much lower value, and therefore, it is recommended that the editorial office carefully consider accepting.

Readability issue: I still fail to get a clear description of why dandelions were chosen for the study model from the Introduction. Although it is mentioned in section 4.2, it does not follow the logical order of reading. If the formatting requirements of the journal dictate that the Materials and Methods be placed after the Results and Discussion, then the article needs to be written in such a way that the important underlying information is written in a reasonable manner at the beginning of the Results. I didn't see such a formulation causing me to spend a long time trying to understand what the article was presenting. For example, all data were collected at six locations A~F, but the figure notes contain only the names, including the text about the data without any details of these six locations or “see Figure 12”.

Author Response

Ad Review

Thank you for your review. As for the allegation that the data collected is from several years ago (3-4 years), we are studying a phenomenon and not a specific year; it is difficult to assume that the response of plants to pollution changes year to year (except for intensity).

We agree with the reviewer that the method used in this article is a mature technical method, but from a scientific point of view, the power of this work is due precisely to the use of the EPR method in the study of EPRF radicals. We would like to emphasize again that the study of EPRF content using the EPR method is not common in the literature (in contrast to studies using spin traps). There are papers in the literature describing the antioxidant capacity of various parts of plants, but not on measuring the persistent radicals mediated by EPFR.

Ad Readability issue:

The Introduction was completed with a paragraph explaining our choice of dandelions. For more details on this topic, see Section 4.2, which, in accordance with the journal's preferred format, is located at the end of the work, after the Results and Discussion section and before the Conclusions.

Added paragraph: “In studying the plant's response to environmental stress, it is important to select an appropriate bioindicator. Dandelion is a plant often used for biomonitoring (Bretzel et al. 2014, Chemerys et al. 2020) because it meets the necessary criteria. It has a broad ecological optimum, is common and easy to identify and obtain, is resistant to contaminants, shows measurable responses at different sites, and with the same exposure to contaminants, should show similar and specific responses (Keane et al. 2001, Degórska 2015, Ianovici 2016).”

Reviewer 2 Report

Comments and Suggestions for Authors

Accept

Author Response

Thank you for your review.